# Cell wall *O*-acetyl and methyl esterification patterns of leaves reflected in atmospheric emission signatures of acetic acid and methanol

Rebecca A. Dewhirst[1]*, Cassandra A. Afseth[1¤], Cristina Castanha[1], Jenny C. Mortimer[2,3], Kolby J. Jardine[1]

1 Climate and Ecosystem Sciences Division, Lawrence Berkeley National Laboratory, Berkeley, CA, United States of America, 2 Joint BioEnergy Institute, Emeryville, CA, United States of America, 3 Environmental Genomics and Systems Biology, Biosciences Division, Lawrence Berkeley National Laboratory, Berkeley, CA, United States of America

¤ Current address: School of Integrative Biology, University of Illinois, Urbana-Champaign, IL, United States of America

* radewhirst@lbl.gov

**Data Availability Statement:** All associated data files are available from Mendeley Data accessed at http://dx.doi.org/10.17632/7bdwbwy6wn.1

## Abstract

Plants emit high rates of methanol (meOH), generally assumed to derive from pectin demethylation, and this increases during abiotic stress. In contrast, less is known about the emission and source of acetic acid (AA). In this study, *Populus trichocarpa* (California poplar) leaves in different developmental stages were desiccated and quantified for total meOH and AA emissions together with bulk cell wall acetylation and methylation content. While young leaves showed high emissions of meOH (140 μmol m$^{-2}$) and AA (42 μmol m$^{-2}$), emissions were reduced in mature (meOH: 69%, AA: 60%) and old (meOH: 83%, AA: 76%) leaves. In contrast, the ratio of AA/meOH emissions increased with leaf development (young: 35%, mature: 43%, old: 82%), mimicking the pattern of *O*-acetyl/methyl ester ratios of leaf bulk cell walls (young: 35%, mature: 38%, old: 51%), which is driven by an increase in *O*-acetyl and decrease in methyl ester content with age. The results are consistent with meOH and AA emission sources from cell wall de-esterification, with young expanding tissues producing highly methylated pectin that is progressively demethyl-esterified. We highlight the quantification of AA/meOH emission ratios as a potential tool for rapid phenotype screening of structural carbohydrate esterification patterns.

## Introduction

Plant cell walls are highly complex structures largely composed of polysaccharides such as cellulose [1], hemicellulose [2], and pectin [3], that account for the majority of plant biomass. Cell walls provide the shape, strength and flexibility needed for numerous physiological processes including cell adhesion and expansion, intercellular communication, and defense against abiotic and biotic stress [4]. The dynamic nature of cell wall response is facilitated by

**Funding:** This material is based upon work supported by the U.S. Department of Energy (DOE), Office of Science, Office of Biological and Environmental Research (BER), Biological System Science Division (BSSD), Early Career Research Program under Award number FP00007421 to Lawrence Berkeley National Laboratory. This work was also supported as part of the DOE Joint BioEnergy Institute (http://www.jbei.org) supported by the U. S. DOE, BER, through contract DE-AC02-05CH11231 between Lawrence Berkeley National Laboratory and the US Department of Energy. This material is based upon work supported by the U.S. DOE, BER, Next-Generation Ecosystem Experiments–Tropics Project (NGEE-Tropics), of the U.S. Department of Energy under contract No. DE-AC02-05CH11231 The funders had no role in study design, data collection and analysis, decision to publish, or preparation of the manuscript.

**Competing interests:** The authors have declared that no competing interests exist.

**Abbreviations:** AA, acetic acid; AIR, alcohol insoluble residue; GC-MS, gas chromatography mass spectrometry; meOH, methanol; PTR-MS, proton transfer reaction mass spectrometry.

chemical modifications that can significantly alter physiochemical, mechanical, and biological properties. For example, many cell wall polysaccharides in higher plants can be heavily *O*-acetylated [5,6] and methylated [7] via ester bonds. Although little is known about the biochemical and physiological functions of those cell wall modifications in trees, recent evidence suggests that they are highly dynamic and play central roles in the control of cell wall growth and tissue development [8], facilitate within and between plant signaling in response to abiotic and biotic stress [9–12], and integrate into primary $C_1$ and $C_2$ metabolism [13]. Moreover, studies in *Arabidopsis thaliana* have highlighted the critical roles cell wall esterification and de-esterification play in the proper development and functioning of xylem vessels [14] and leaf stomata [15]. A lack of xylan *O*-acetylation resulted in collapsed xylem vessels which greatly altered plant water use [14] through a reduction in xylan–cellulose interactions [16]. Moreover, pectin de-methylesterification was shown to modify cell wall elasticity and growth rates [17,18], and transgenic plants with guard cell walls enriched in methyl-esterified pectin showed a decreased dynamic range of stomatal conductance with reduced evaporative cooling and growth [15]. The pattern and degree of pectin methylation also impacts plant susceptibility to microbial infection [19], for instance wheat cultivars with more blockwise distribution of methyl esters were more susceptible to fungal infection than cultivars with more random methylation patterns [20]. Therefore, changes in esterification of cell walls and associated transport and metabolism of the released methanol and acetic acid could provide a rapid mechanism for plants to respond to abiotic and biotic stress.

Many land-atmosphere flux studies above agricultural crops [21] and fruit plantations [22] as well as temperate [23,24], boreal [25], and tropical [26] forests have identified meOH as a major, sometimes dominant, component of ecosystem volatile emissions. MeOH production in plants is largely attributed to changes in chemical and physical cell wall properties associated with the hydrolysis of methyl esters of cell wall carbohydrates like pectin [11,27–29]. However, this assertion lacks experimental evidence, which we aim to address in the present study. Foliar meOH emissions are tightly associated with growth, abiotic and biotic stress, and senescence processes and are generally attributed to pectin de-methylation reactions, including the action of pectin methylesterases [30], associated with physicochemical changes in cell walls [31–33]. For example, foliar meOH emissions tightly correlate with leaf expansion rates [34] and numerous studies have shown that young expanding leaves emit greater amounts of meOH than mature leaves [27,34,35]. Moreover, foliar meOH emissions are highly sensitive to leaf temperature with a factor of 2.4 increase in emissions reported for each 10°C increase in leaf temperature [36]. Pectin methylesterases are activated during plant pathogen penetration of leaf cell walls, producing methanol [12]. This methanol is thought to have a priming role in leaves, increasing resistance to bacteria but increasing sensitivity to viruses in *Nicotiana benthamiana* [37].

In contrast to meOH, relatively few studies have reported plant acetic acid (AA) emissions and little information is available regarding its biochemical source(s). A recent study observed that during leaf senescence, both meOH and AA emissions were simultaneously stimulated [38]. Although acetate is a known product of *O*-acetylation hydrolysis of cell walls, connections between plant AA emissions and cell wall *O*-acetyl hydrolysis have not been investigated. Therefore, quantitative evidence linking changes in cell wall esterification and plant-atmosphere emissions of meOH and AA is lacking. Nonetheless, meOH and AA emissions from managed and natural ecosystems can be expected to increase with climate warming [39] and increased forest turnover rates associated with land use change including biomass burning [40], increased abiotic and biotic stress [41], and secondary forest regeneration through the release of suppressed trees and increased pioneer species recruitment rates [42]. Therefore, it is vital to quantify the relationships between cell wall esters and foliar meOH and AA

emissions to evaluate the hypothesis that emissions derive from cell wall de-esterification and to understand their physiological and biochemical roles during plant growth and development, adaptation to abiotic and biotic stress, and mortality and decomposition.

Cell walls are widely used as a renewable feedstock source for the production of biofuels and bioproducts [43]. However, *O*-acetylation [6] of cell walls can compromise microbial fermentation yields. Acetate released during biomass processing can accumulate to concentrations higher than 10 g/L in cellulosic hydrolysates leading to the inhibition of ethanol production by some organisms, including *Saccharomyces cerevisiae*, the principal microorganism used to produce ethanol [44]. Economic models estimate that a 20% reduction in biomass *O*-acetylation could result in a 10% reduction in ethanol price [45]. However, engineering of the cell wall in wheat and tobacco to reduce pectin demethylation improved saccharification of plant tissues by 30–40% [46]. Thus, from a biotechnological point of view, *O*-acetylation and methyl esterification of cell wall polysaccharides impacts the efficiency of their conversion to ethanol in a complex manner by both inhibiting fermentation and enhancing saccharification. Therefore, the development of rapid, non-destructive tools to quantify cell wall esterification patterns in bioenergy plants is of high interest.

In this study, the economically, environmentally, and ecologically important California poplar (*Populus trichocarpa*), a tree species with emerging potential for use as a biofuel [47], was utilized to characterize quantitative relationships between meOH and AA emissions and bulk cell wall *O*-acetylation and methylation patterns. We hypothesized that the main biochemical source of foliar emissions of meOH and AA is cell wall de-esterification. Given previous observations of decreasing foliar meOH emissions with leaf age [34], and the de-esterification of cell walls throughout plant development [48,49] we hypothesize that similar phenological pattern can be observed for AA emissions. By normalizing *O*-acetyl ester content of isolated leaf cell walls with methyl ester content, this hypothesis also predicts that foliar emissions will reflect the cell wall *O*-acetyl/methyl ester 'signature'.

## Materials and methods

In this study, we quantified real-time and total meOH and AA emissions and water loss from detached poplar leaves undergoing desiccation together with bulk cell wall *O*-acetyl and methyl ester content of leaves in three stages of development (Fig 1).

### Plant material

California poplar (*Populus trichocarpa*) trees were obtained from Plants of the Wild (Washington State, USA) and maintained outdoors at the Oxford Tract Experimental Farm in Berkeley, CA, USA, where they were regularly watered and maintained pest free. During active experimentation, single trees were moved into a growth chamber (Percival Intellus Control System, Iowa, USA) and kept at 27.5°C during the day (5:45 am-8:00 pm; 30% light) and 23°C at night (8:00 pm to 5:45 am).

Seven leaf samples from each age class (young, mature, and old) were used from a total of four individual trees. Leaf age was determined as previously reported [50], with young leaves light green and not fully expanded, mature leaves dark green and fully expanded, and old leaves with the beginnings of brown senescence on the edges.

For each sample, four leaves (two each for emissions and cell wall analysis) of the same age category were harvested from the tree, with leaf area and fresh weight determined for each pair of leaves. The leaves for emissions analysis were immediately placed in the dynamic leaf chamber, and the leaves designated for cell wall ester quantification were immediately flash frozen in liquid nitrogen and stored at -70°C.

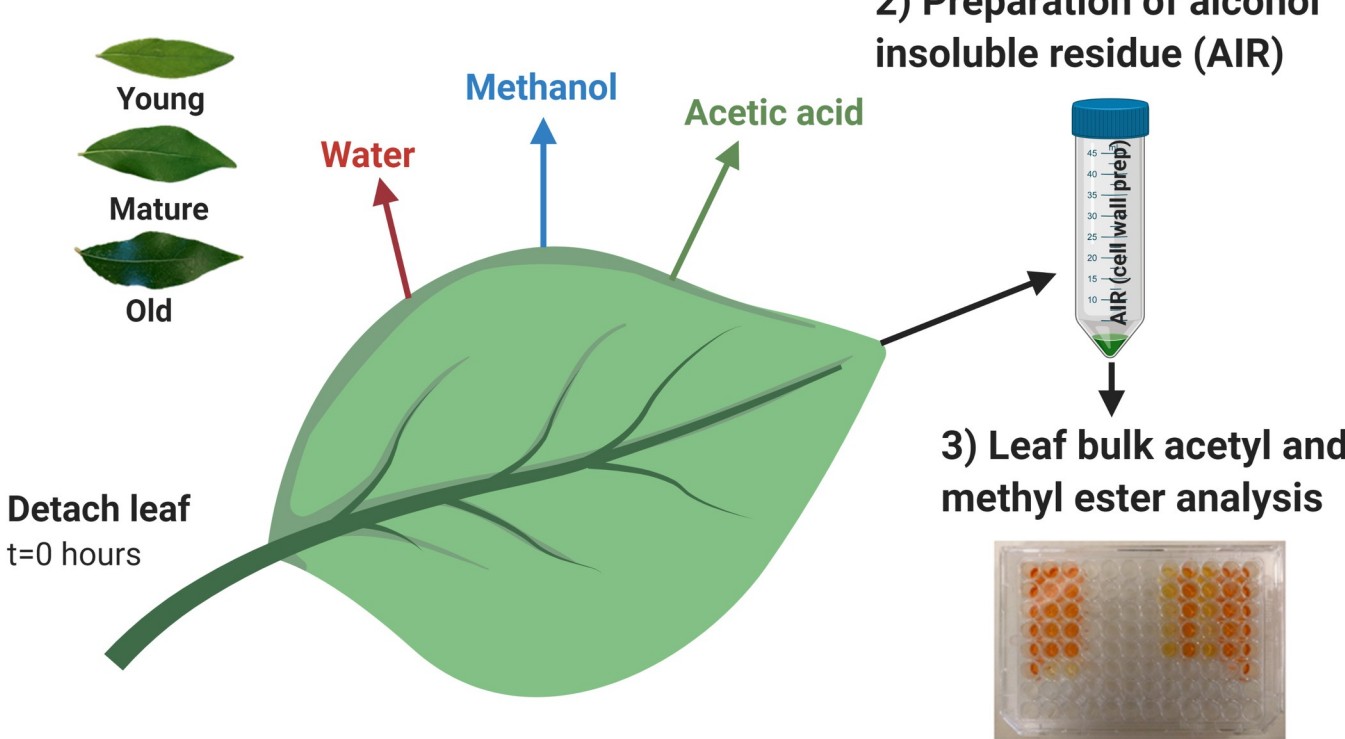

**Fig 1. Overview of experimental design.** Experimental design with coupled gas exchange and cell wall esterification analysis during leaf desiccation experiments.

### Dynamic leaf chamber

A 475-ml glass chamber with a flow-through of 300 ml min$^{-1}$ of dry hydrocarbon-free air exposed to 1000–1500 μmol m$^{-2}$ s$^{-1}$ photosynthetic photon flux density was used to desiccate detached poplar leaves and quantify real-time (nmol m$^{-2}$ s$^{-1}$) and total emissions (nmol m$^2$) of meOH and AA. A tee downstream of the chamber diverted 75 ml min$^{-1}$ of the air exiting the chamber to the proton transfer reaction mass spectrometer (PTR-MS, described below) and 25 ml min$^{-1}$ to the gas chromatography mass spectrometer (GC-MS) when sampling. Excess air was vented to the room via a second tee. Background measurements of volatile concentrations from the empty chamber were collected for approximately two hours by both analytical systems prior to the introduction of the leaves. Following the addition of the two leaves to the chamber, emissions were continuously quantified for 16.7 hours. The total amount of volatiles emitted per m$^2$ of leaf was calculated by integrating the emission curve across the 16.7 hours.

### Online PTR-MS and GC-MS

Gaseous samples containing meOH and AA from the desiccation chamber were collected and dehydrated with an air server interfaced with a Kori-xr dehumidifier coupled to a Unity-xr thermal desorption system (Markes International, UK). Air samples (25 mL/min x 10 min: 0.25 L) were first dehydrated by passing the air sample through the Kori-xr held at -20 °C before the volatiles were pre-concentrated onto the cold trap (Air toxics, Markes International, UK) held at -30 °C with the sample flow path maintained at 150 °C. The collected meOH and

AA were subsequently quantified by GC-MS by injection onto a capillary column (Rtx-VMS, 60 m x 0.25 mm x 1.4 μm) interfaced with a gas chromatograph (7890B, Agilent Technologies, CA, USA) with a high efficiency source electron impact quadrupole mass spectrometer (5977B HES MSD, Agilent Technologies, CA, USA). During injection of the sample onto the analytical column, the cold trap was rapidly heated to 280 °C for three minutes while back-flushing with carrier gas at a flow of 6.5 mL/min. In order to improve peak shape and further reduce the amount of water introduced into the GC-MS, 5 mL/min of this flow was vented through the split while the remaining 1.5 mL/min was directed to the column, temperature programmed with an initial hold at 40°C for 1.5 min followed by an increase to 170°C at 15°C min$^{-1}$. A post run temperature of 230°C was applied for 1.5 min. The mass spectrometer was configured for trace analysis (SIM Mode and 10 X detector gain factor) with 50 ms dwell times for the target compounds; methanol (m/z 31, 29, 15) and acetic acid (m/z 43, 45, 60). Quantification of the volatile concentrations was based on linear calibration curves of a primary gas standard (Restek Corporation, PA, USA). Calibration curves were generated for m/z 31 (meOH, retention time 6.0 min) and m/z 60 (AA, retention time 9.4 min) for 0.0, 2.3, 4.6, 6.9, 9.1 and 11.3 nL of the collected gas primary standard. The online GC-MS was programmed to automatically collect and analyze 5 sequential samples from the empty chamber, followed by 40 samples with the two sample leaves inside the chamber (measurement frequency 27–30 min).

In parallel with the GC-MS, quantification of meOH and AA gas-phase concentrations exiting the leaf chamber were made in real-time using a high sensitivity quadrupole proton transfer reaction mass spectrometry (PTR-MS, Ionicon, Austria, with a QMZ 422 quadrupole, Balzers, Switzerland). The PTR-MS was operated with a drift tube voltage of 600 V and pressure of 1.9 mb. The following mass to charge ratios (m/z) were sequentially monitored during each PTR-MS measurement cycle: m/z 32 ($O_2^+$) and m/z 37 ($H_2O$-$H_3O$+) with a dwell time of 10 ms, m/z 21 ($H_3{}^{18}O^+$) with a dwell time of 50 ms and m/z 25 (dark counts), m/z 33 (methanol), m/z 43 (acetate fragment) and m/z 61 (acetic acid) with a dwell time of 5 s each. Quantification of the volatile concentrations was based on linear calibration curves of a primary gas standard (Restek Corporation, USA). Calibration curves were generated for m/z 33 (methanol) and m/z 61 (acetic acid) for 0.0, 9.4, 18.5, 27.5, 36.4 and 45.0 ppb of the gas primary standard.

## Preparation of leaf whole cell wall samples

Alcohol insoluble residue (AIR; composition dominated by whole cell wall material) of each leaf sample was prepared as previously described [51]. Briefly, leaf samples were flash frozen in liquid nitrogen and stored at -70°C before AIR preparation. The leaves were ground using a mortar and pestle on dry ice to avoid thawing, and the powder was then incubated in 70°C ethanol (96% v/v) for 30 minutes. The samples were centrifuged (Eppendorf Centrifuge 5417R, Germany) and the pellet was washed sequentially in 96% ethanol, 100% ethanol, twice in methanol:chloroform (2:3 v/v), 100% ethanol, 65% ethanol, 80% ethanol and 100% ethanol. The samples were incubated for 1 hour with shaking (1000 rpm) in each of the methanol:chloroform steps. After the final wash the samples were dried in a speedvac (Eppendorf Vacufuge Plus, Germany) at 30°C.

## Bulk methyl and O-acetyl ester quantification in AIR samples

Bulk methyl and *O*-acetyl ester content of AIR samples was carried out using commercial kits (Methanol Assay Kit, and Acetate Assay Kit, BioVision, CA, USA). AIR samples (2.5 mg) were saponified with NaOH (1 M, 125 μL) for 16 hours then neutralized with 1 M HCl. The samples were centrifuged (10 minutes at 15000 rpm) and 5 μL of the supernatant was transferred to a 96-well plate. The samples were treated with the assay kit enzymes and plates incubated at

37˚C for 30 minutes (for methanol) or at room temperature for 40 mins (for acetate). Absorbances were measured at 450 nm (for both assays) on a 96-well plate reader (SpectraMax M2; Molecular Devices, CA, USA). Total methyl and *O*-acetyl content of the AIR samples (µg/mg AIR) were determined by including a six-point calibration on each plate using the included standard).

## Statistical analysis

Statistically significant differences in AA and meOH emissions and leaf bulk cell wall ester content between leaf developmental categories were assessed by a one-way analysis of variance (one-way ANOVA) and a Tukey's post hoc test to evaluate significant differences between the means. The same statistical analysis was carried out on AA/meOH ratios from leaf emissions and AIR with all analysis carried out in R version 3.6.0.

## Results

### Methanol and acetic acid emissions as detected by online PTR-MS and GC-MS

A coupled PTR-MS and online GC-MS system was applied to the quantification of meOH and AA from desiccating leaf samples placed in an illuminated glass chamber with hydrocarbon free air flowing through. In the absence of a leaf, the empty chamber showed low background concentrations for meOH and AA of < 0.1 ppb, which greatly increased upon introducing the leaf to the chamber. For all leaf samples, meOH and AA emissions initially resulted in large peaks, typically lasting 20–40 min, that tapered off as the leaf dried (with humidity monitored qualitatively by m/z 37) (Fig 2).

Because leaf emissions initially changed quickly, whereas GC-MS collection occurs slowly (e.g. samples collected for 10 min every 30 min), the GC-MS typically underestimated the magnitude of emissions during this initial period relative to the PTR-MS, which has a 22 sec cycle time (e.g. Fig 2A and 2B). This is related to the fact that while the PTR-MS continuously monitored emissions following the introduction of the leaves into the chamber, the GC-MS had a delay of a few minutes while it prepared to collect the first sample, during which time a large proportion of the initial emissions had already occurred. However, over the remaining period, where emissions were lower but changed more slowly, a good quantitative comparison of the emission rates between PTR-MS and GC-MS was generally observed. AA, and to a lesser extent meOH, showed a second large peak in emissions several hours after the leaves were introduced, coinciding with a rapid decrease in chamber air humidity. As AA is highly water soluble, we attribute this trend observed in leaves of all developmental stages, to condensation of water inside the chamber from the initial leaf transpiration, dissolving leaf-derived AA into the water, followed by evaporation and release of gaseous AA associated with a rapid drop in chamber humidity following leaf desiccation. Despite issues with condensation, by integrating meOH and AA emissions throughout the entire 16.2 hour experiments, an accurate quantification of total meOH and AA was obtained. Following the drying of the chamber humidity and loss of water from the leaf, a small increase in meOH emissions (and sometimes AA) was observed (e.g. after 10 hr in Fig 2).

### MeOH and AA emissions decrease with increasing leaf age

For each of the leaf developmental stages, mean total emissions of meOH and AA were quantified (Fig 3A and 3B). Young leaves showed strong emissions of meOH (140 µmol m$^{-2}$) and AA (42 µmol m$^{-2}$), while emissions were reduced in mature (meOH: 69%, AA: 60%) and old

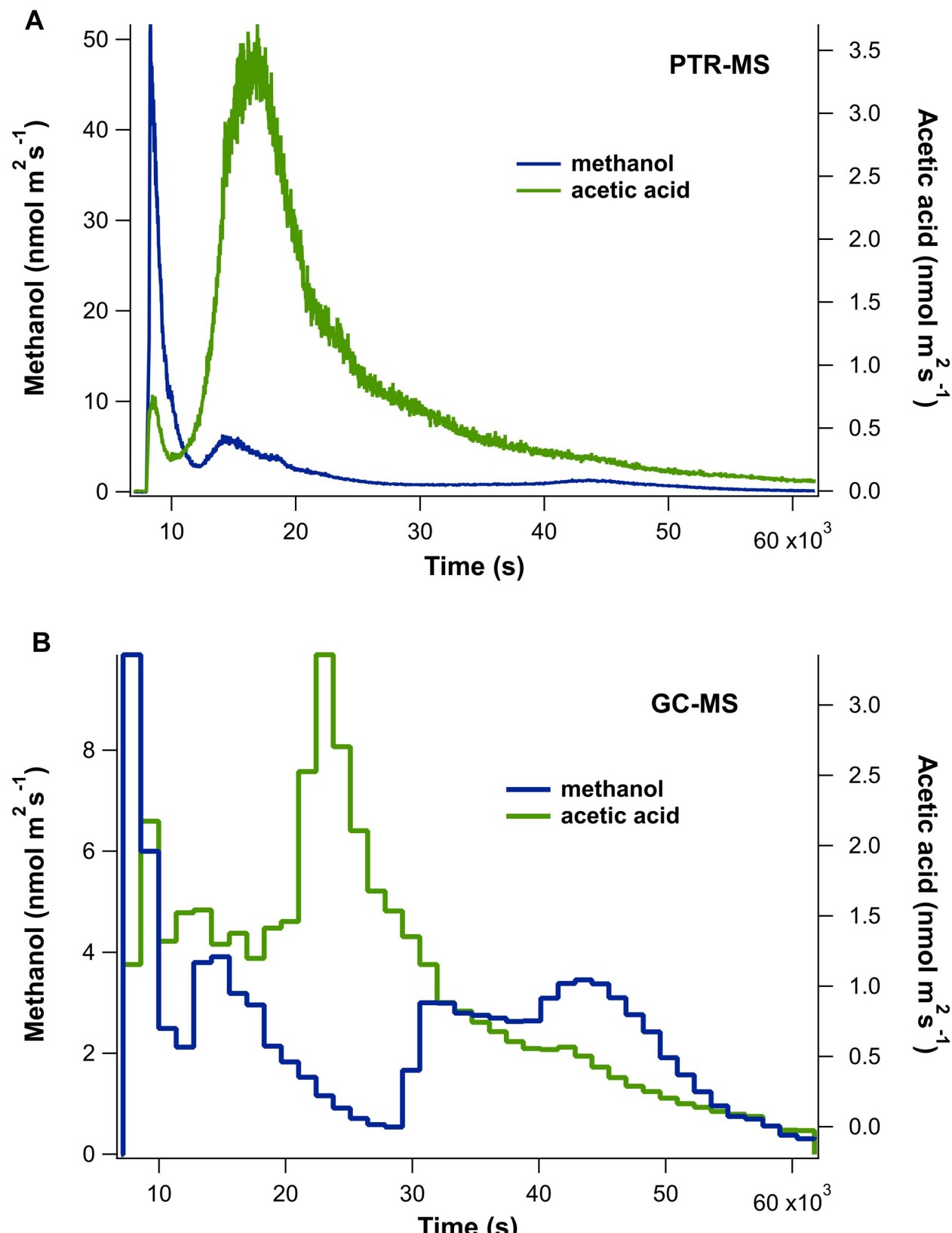

**Fig 2. Leaf emissions of AA and meOH during desiccation.** Example real-time leaf emissions of acetic acid (AA) and methanol (meOH) using simultaneous analysis by **A**) PTR-MS and **B**) online TD-GC-MS during a 16.7 hour (60,000 s) desiccation experiment.

(meOH: 83%, AA: 76%) leaves. The difference in emissions of both meOH and AA across the developmental stages were statistically significant. Specifically, the differences between young and mature ($p = 0.0012$ for acetate and $p = 0.0053$ for methanol, ANOVA and Tukey post hoc analysis) and between young and old leaf emissions were statistically significant ($p < 0.001$, ANOVA and Tukey's post hoc test, for both compounds). It is possible that like meOH, AA emissions also derived from the de-esterification of cell wall esters. To further investigate this possibility, we normalized AA emissions by meOH emissions and compared these 'signatures' to those from leaf cell wall methyl and *O*-acetyl esters.

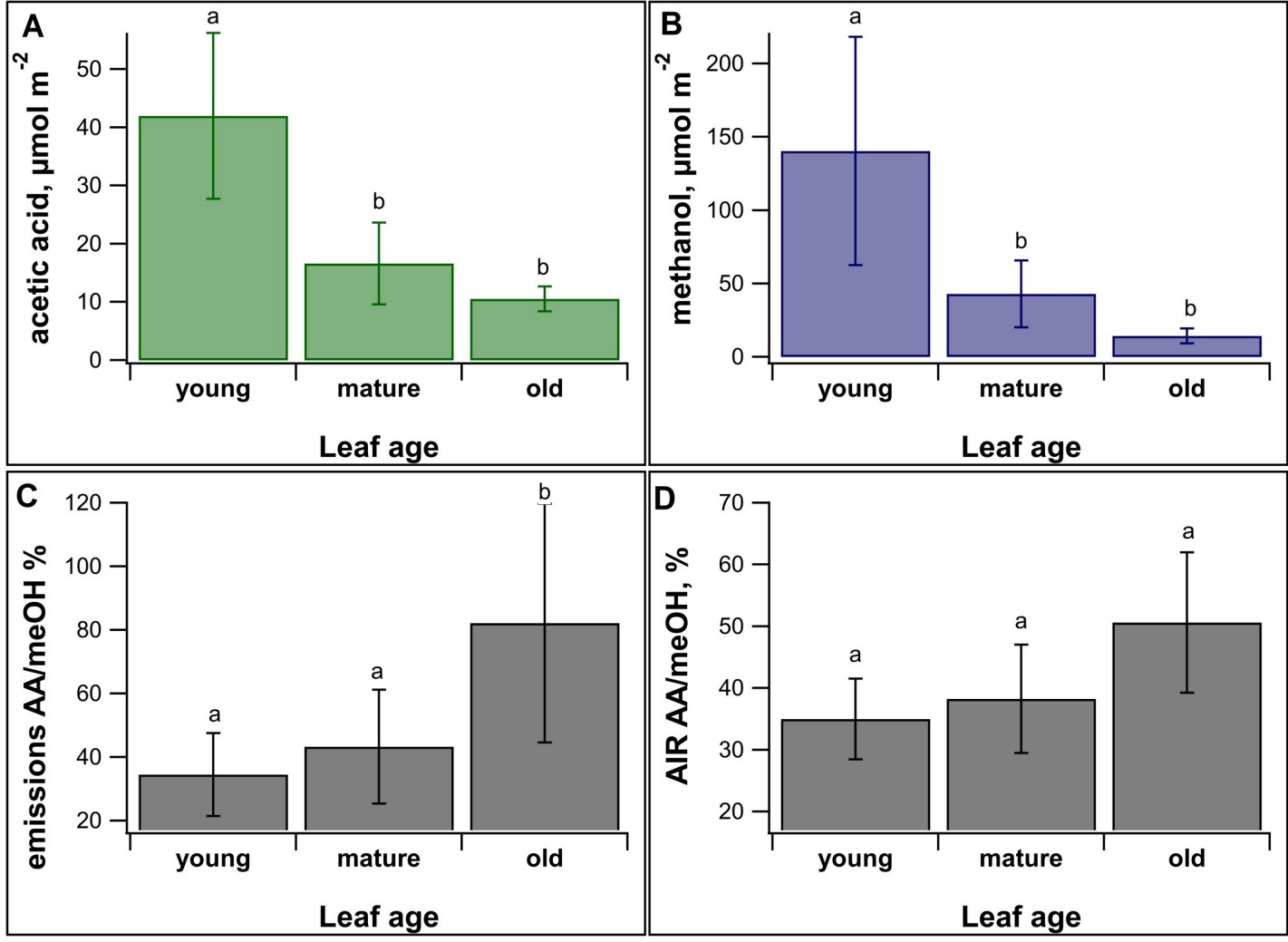

**Fig 3. Total emissions and cell wall esters across leaf developmental stages.** Average total leaf emissions of **A.** acetic acid (AA) and **B.** methanol (meOH), among 3 leaf developmental stages (young, mature, and old), were quantified using PTR-MS. Also shown as a function of leaf developmental stage are **C.** AA/meOH leaf emission ratios and **D.** AA/meOH ratios of saponified whole leaf cell wall preparations (AIR). Error bars represent +/- one standard deviation (n = 7 leaves for each age class). Statistically significant (p<0.01; ANOVA and Tukey post hoc analysis, n = 7) differences are indicated with different letters.

### AA/meOH emissions ratios are reflected by cell wall O-acetyl/methyl ratios

With leaf age, while the absolute emission rates decreased, the ratio of AA to meOH emissions increased (Fig 3C). Moreover, the AA/meOH emission signatures showed similar magnitudes and dependence on leaf developmental stage as the O-acetyl/methyl ratios of the isolated leaf cell walls (Figs 3D and 4). Young leaves had an emissions ratio of 34.5 ± 13.0% and cell wall ester ratio of 35.0 ± 6.5%, mature leaves had emission ratios of 43.3 ± 17.9% and cell wall ester ratio of 38.2 ± 8.8%, and old leaves had an emission ratio of 82.2 ± 37.6% and a cell wall ester ratio of 50.7 ± 11.4%. The difference in AA/meOH emissions ratio across developmental stages was statistically significant (p = 0.0084 between young and old, and p = 0.027 between old and mature, ANOVA and Tukey's post hoc analysis). Across all the age categories AA/meOH emissions ratios were correlated with AA/meOH cell wall ester content with an $R^2$ value of 0.4 (Fig 4A). This $R^2$ value increase to 0.99 when just the averages of each age category are considered (Fig 4B). These results are consistent with meOH and AA emission sources from cell wall de-esterification of both pectin and hemicelluloses. The increase in emission ratios with age was driven by an increase in total O-acetyl ester content and corresponding decrease in methyl ester content of cell walls (Fig 5). Old leaves had statistically higher levels of acetate esters (p = 0.045, ANOVA and Tukey's post hoc analysis) than young leaves.

## Discussion

Previous studies on volatile plant emissions rarely include both meOH and AA due to technical difficulty in quantifying low ppb concentrations of these compounds in high humidity air samples. While fast online techniques like PTR-MS are increasingly used to study plant volatile emissions, their accurate calibration with primary standards are often neglected. Moreover, validation of time-series using GC methods are rarely performed due to the high water solubility of AA, causing analytical losses when air samples are dehydrated, and its 'sticky' nature that often shows strong memory effects and even losses due to strong gas-surface interactions. Here we overcome these limitations and demonstrate for the first time the reliable and robust identification and quantification of both meOH and AA emissions using a coupled PTR-MS and online GC-MS system, optimized for high specificity and sensitivity to ppb concentrations of meOH and AA in humid air samples. Key to the direct quantification of trace AA emission by GC-MS was the high sensitivity of the GC-MS, which allowed analysis of low volume (250 mL) air samples, and dehydration of the air sample by passing it through inert tubing at -20°C, before quantitatively trapping the AA at -30°C on an activated carbon sorbent. Regular calibrations of the PTR-MS and GC-MS to a primary gas standard showed high linearity ($R^2$ = 0.95–0.99) and sensitivity to meOH and AA (S1 Fig).

In this study, we observed two distinct emissions of meOH and sometimes AA, an initial emission as the leaves were starting to dry out, and a second smaller peak that occurred when the leaf was considered dry (e.g. the water vapor concentration in the leaf chamber returned to the value of the empty chamber with dry clean air passing through). This is consistent with what has been observed from cut grass and clover: A burst of meOH emissions due to cutting the leaves and stems followed by a second emission lasting for several hours when the vegetation was starting to dry out [52].

The observations that meOH emissions are reduced in mature leaves compared to young leaves are consistent with a number of studies showing a decrease in leaf meOH emissions with leaf age [27,34,35] including a recent study that demonstrated the fading of meOH emissions during *Populus tremula* leaf maturation [53]. While AA emissions throughout leaf development have been little studied, our observation that AA emissions followed the same

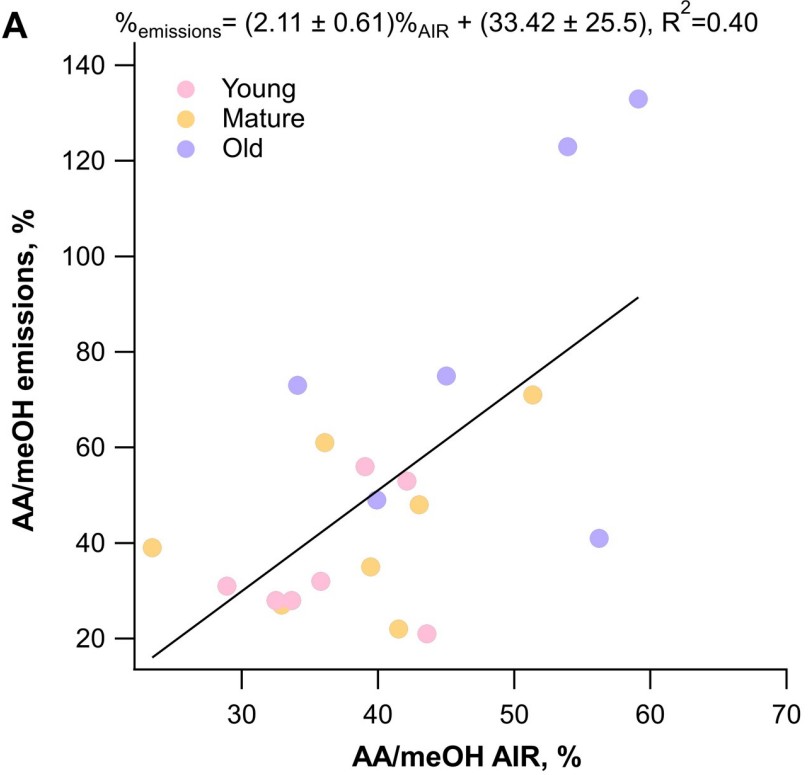

**A** $\%_{\text{emissions}} = (2.11 \pm 0.61)\%_{\text{AIR}} + (33.42 \pm 25.5)$, $R^2 = 0.40$

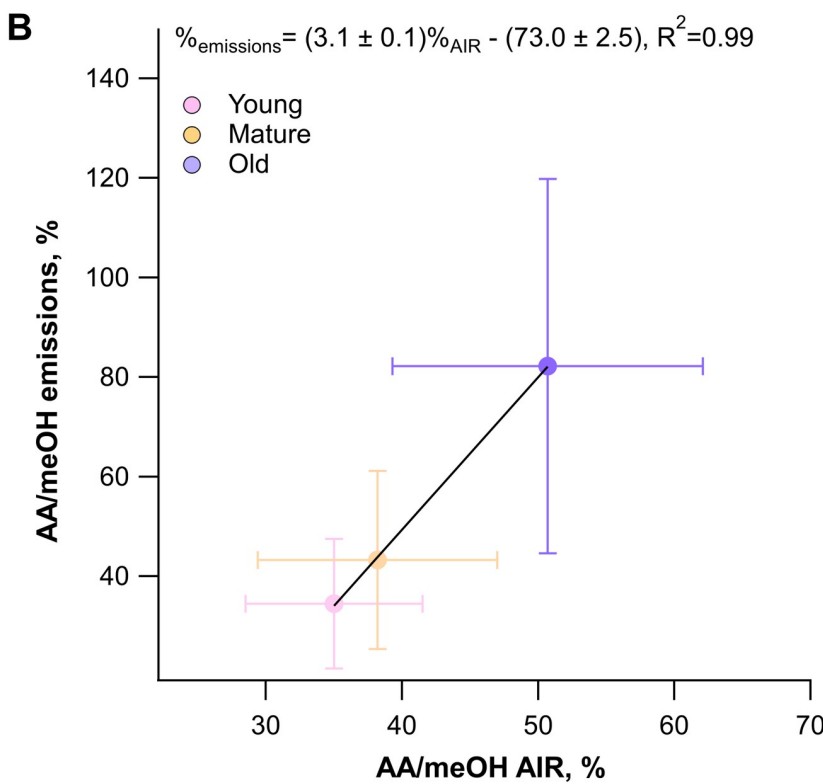

**B** $\%_{\text{emissions}} = (3.1 \pm 0.1)\%_{\text{AIR}} - (73.0 \pm 2.5)$, $R^2 = 0.99$

**Fig 4. Correlation between AA/meOH ratios of emissions and cell wall ester contents.** The AA/meOH ratio for emissions was determined by PTR-MS and the cell wall ester contents were quantified using colorimetric assay kits on saponified AIR samples. The linear equations describing the relationship between emissions and cell wall ratios is shown ± standard deviation, along with the $R^2$ value for all the data (**A**) and for the average of each leaf age category (**B**).

phenological pattern as meOH emissions, supports the hypothesis that, like meOH, AA emissions derive from de-esterification of cell wall esters.

The observed increase in cell wall $O$-acetylation coupled with a decrease in methylation are similar to those reported in winter oil flax where $O$-acetylation of acid-soluble pectins increased throughout leaf development (23–40%) [54]. The observations are consistent with the emerging view that during cell wall biogenesis, the composition and corresponding architecture of the wall changes, which may impact the $O$-acetyl/methyl ratios. The matrix polysaccharide of primary cell wall pectin is partially replaced by hemicelluloses in the secondary cell wall that can have higher $O$-acetyl acetate content due to the high levels of $O$-acetyl-(4-$O$-methylglucurono)-xylan [55,56] (Fig 6A). Moreover, young growing leaves, where new cell walls are being synthesized, are enriched in highly methylated [48] pectin (Fig 6B) that is purportedly progressively demethyl-esterified throughout cell expansion and aging. For example, newly synthesized homogalacturonan is transported to the cell wall with a high degree of methylation. The methyl groups are then hydrolyzed by pectin methylesterases releasing meOH under tight spatial and temporal control during development [8]. In contrast, pectin can be both acetylated and deacetylated *in muro* by pectin acetylesterases [57]. Therefore, determining the location of $O$-acetyl and methyl groups on specific polysaccharides is a valuable next step in this study, which will allow us to test the hypothesis that leaf cell wall $O$-acetyl content increases throughout development due to an increase in total xylan content, and therefore total bulk cell wall $O$-acetylation. Moreover, the hypothesis that cumulative demethylation

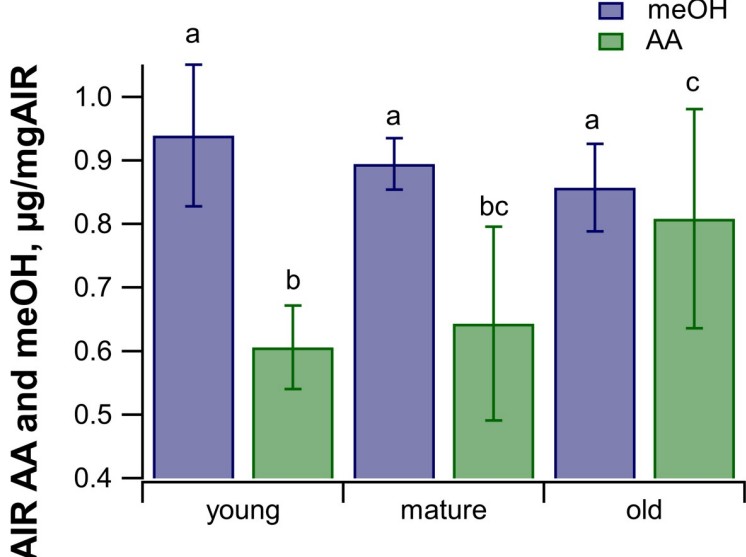

**Fig 5. Average cell wall acetate and methanol content for each leaf age category.** Cell wall acetate and methanol were quantified from saponified AIR for each age category using colorimetric assay kits. Error bars represent ± one standard deviation (n = 7 for each age category). Statistically significant (p<0.05; ANOVA and Tukey post hoc analysis, n = 7) differences are indicated with different letters.

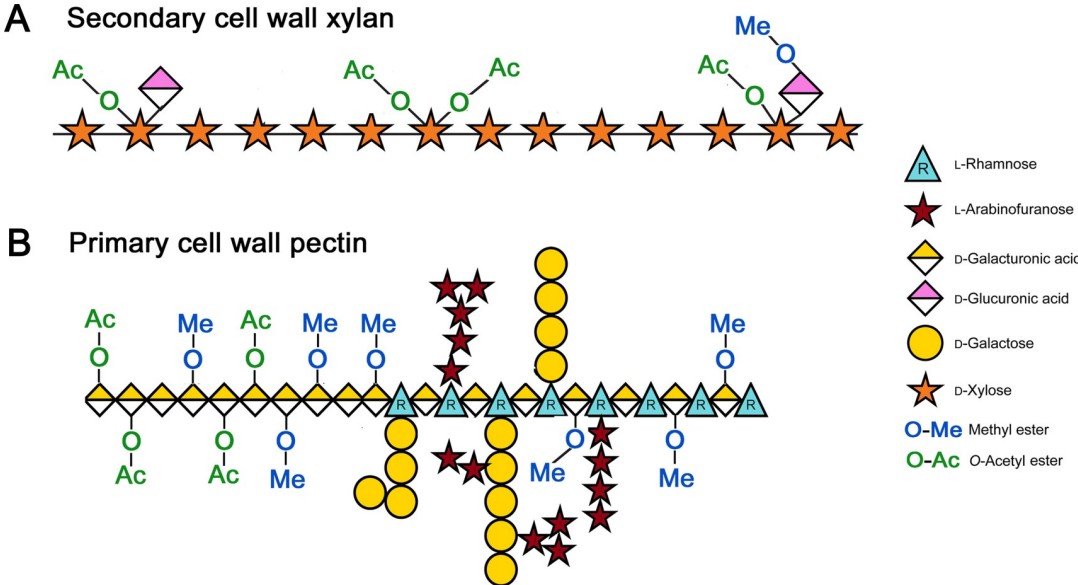

**Fig 6. Structure of *O*-acetylated and methylated cell wall polysaccharides.** Structures of acetylated xylan (**A**) and acetylated and methylated pectin (**B**) are shown.

of pectin during leaf development accounts for the decrease in bulk leaf cell wall methyl content during aging could also be tested.

Currently the only methods for measuring *O*-acetyl and methyl ester content of plant cell walls are destructive, involving costly and time-consuming techniques of harvesting plant tissues, isolating cell walls, and conducting separate analysis of cell wall methyl and *O*-acetyl esters. In this study, we showed that *P. trichocarpa* leaf cell wall *O*-acetyl/methyl ester ratios, and their dependence on leaf developmental stage, were quantitatively reflected in the AA/meOH emission ratio during leaf desiccation, providing evidence for cell wall esters as the source of biogenic meOH and AA. We therefore suggest quantifying AA/meOH emission ratios may present a new non-destructive tool to study esterification in plant cell walls at various spatial (leaf to ecosystem) and temporal (minutes to seasons) scales. As esterification of plant cell walls can have a large impact on saccharification and fermentation of plant biomass, while influencing plant physiology, quantification of AA/meOH emission ratio may present a new method for rapid phenotype screening of cell wall ester composition of plants. In the near future, it will be necessary to grow dedicated bioenergy crops as a feedstock for the production of liquid transport fuels and bioproducts. The presented methods will help advance rapid phenotype screening and genetic manipulation of the cell wall ester content, with the goal of increasing biofuel yields and plant resistance to abiotic stress. Moreover, the methods can be used in future studies to help understand the impacts of cell wall esterification on cell wall structure and function, and numerous physiological and biochemical process including growth and development, stress responses and signaling, plant hydraulics, and central carbon metabolism. Therefore, in situ monitoring of atmospheric emissions of meOH and AA from terrestrial ecosystems could help improve predictions of both tree growth and mortality mechanisms and their sensitivities to environmental change.

## Supporting information

**S1 Fig. PTR-MS and online GC-MS calibration.** Example linear calibration responses for PTR-MS (**A**-**B**) and online GC-MS (**C**-**D**) to a primary gas-phase standard of acetic acid (AA)

and methanol (meOH) on 22 June 2019.
(TIF)

**S1 Dataset. Raw experimental data files.**
(DOCX)

## Acknowledgments

We would like to kindly acknowledge Christina M. Wistrom in the UC Berkeley Oxford Tract greenhouse for the support in growing and maintaining the commercial poplar trees used in this study. Fig 1 was created with Biorender.com.

## Author Contributions

**Investigation:** Rebecca A. Dewhirst, Cassandra A. Afseth, Kolby J. Jardine.

**Methodology:** Rebecca A. Dewhirst, Cristina Castanha, Jenny C. Mortimer, Kolby J. Jardine.

**Resources:** Cristina Castanha.

**Supervision:** Cristina Castanha, Jenny C. Mortimer, Kolby J. Jardine.

**Writing – original draft:** Rebecca A. Dewhirst, Cassandra A. Afseth, Jenny C. Mortimer, Kolby J. Jardine.

**Writing – review & editing:** Rebecca A. Dewhirst, Cassandra A. Afseth, Cristina Castanha, Jenny C. Mortimer, Kolby J. Jardine.

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
