## [Decision Letter · Decision Letter 0]

7 Feb 2020

PONE-D-19-34824

Cell wall O-acetyl and methyl esterification patterns of leaves reflected in atmospheric emission signatures of acetic acid and methanol

PLOS ONE

Dear Dr Dewhirst,

Thank you for submitting your manuscript to PLOS ONE. Your manuscript has been peer-reviewed by two experts in the field. As you can see from the comments reported below, both the reviewers found your study very interesting and worthy of publication following some revisions. In short, reviewer #1 asked to expand the ‘Introduction’ and ‘Discussion’ sections to describe more fully the importance of this research work, while reviewer #2 arose some points to be addressed, particularly a more correct presentation of the statistical analysis.

Therefore, we invite you to submit a revised version of the manuscript that includes the reviewers' requests.

We would appreciate receiving your revised manuscript by 07th of March, 2020. To enhance the reproducibility of your results, we recommend that if applicable you deposit your laboratory protocols in protocols.io, where a protocol can be assigned its own identifier (DOI) such that it can be cited independently in the future. For instructions see: http://journals.plos.org/plosone/s/submission-guidelines#loc-laboratory-protocols

We look forward to receiving your revised manuscript.

Kind regards,

Federico Brilli, Ph.D.

Academic Editor

PLOS ONE

Journal Requirements:

2. We note that Figure 1 in your submission contain copyrighted images. All PLOS content is published under the Creative Commons Attribution License (CC BY 4.0), which means that the manuscript, images, and Supporting Information files will be freely available online, and any third party is permitted to access, download, copy, distribute, and use these materials in any way, even commercially, with proper attribution. For more information, see our copyright guidelines: http://journals.plos.org/plosone/s/licenses-and-copyright.

Reviewers' comments:

Reviewer's Responses to Questions

**Comments to the Author**

1. Is the manuscript technically sound, and do the data support the conclusions?

Reviewer #1: Yes

Reviewer #2: Partly

2. Has the statistical analysis been performed appropriately and rigorously? 

Reviewer #1: Yes

Reviewer #2: No

3. Have the authors made all data underlying the findings in their manuscript fully available?

Reviewer #1: No

Reviewer #2: Yes

4. Is the manuscript presented in an intelligible fashion and written in standard English?

Reviewer #1: Yes

Reviewer #2: Yes

5. Review Comments to the Author

Reviewer #1: I very much enjoyed this manuscript and have an unusual recommendation for the authors. I found the Intro and Disc sections well written but somewhat telegraphic. I'm concerned that the significance of the realists may not come across to the general reader. I encourage the authors to consider expanding these sections to discuss the importance of their work at greater length.

Also, I did not see the data link that, I believe, PLOS One requires. If I missed it, I apologize. If it's not yet in the ms., it should be added

Reviewer #2: The manuscript “Cell wall O-acetyl and methyl esterification patterns of leaves reflected in atmospheric emission signatures of acetic acid and methanol” regards to the quantification of total meOH and AA emissions in Populus trichocarpa desiccated leaves at different developmental stages were taking into account the cell wall acetylation and methylation content highlighting the quantification of AA/meOH emission ratios as a potential tool for rapid phenotype screening of structural carbohydrate esterification. I consider the manuscript very interesting and topical in the field of plant physiography and biofuels. Developing a method based on two experimental approaches, able to dynamically quantify acetic acid simultaneously with methanol is very innovative. I find the presentation of the work quite good and I don't find too many problems in the language. However, there are some aspects that need to be reviewed before the work can be presentable.

Comments:

1) The introduction should be reviewed trying to replace some slightly dated references (Micheli et al etc). The role of PME, pectin methylesterification and methanol release in defense against pathogens should be described a little more thoroughly. For references see papers of these authors Bethke, Lionetti, Dorokhov.

2) The sentence on page 5 lane 106 suggests that the AA is expected to shrink during development ... review

3) Should figures 2 in the abscissa axis be seconds or minutes?

4) The statistical treatment of the data is not correctly presented. The histograms are not accompanied by any statistical sign (letters for ANOVA, asterisks, etc.). It is also unclear how many repetitions have been made. Every figure must have adequate references in this regard.

5) Figure 6 is more to review. It must be moved to supplementary materials

6. PLOS authors have the option to publish the peer review history of their article (what does this mean?). If published, this will include your full peer review and any attached files.

Reviewer #1: No

Reviewer #2: No

---

## [Author Response · Author response to Decision Letter 0]

17 Mar 2020

Reviewer 1

Comment 1: I very much enjoyed this manuscript and have an unusual recommendation for the authors. I found the Intro and Disc sections well written but somewhat telegraphic. I'm concerned that the significance of the realists may not come across to the general reader. I encourage the authors to consider expanding these sections to discuss the importance of their work at greater length.

Response 1: We have included more detail on the role of pectin methylesterases and methanol in plant pathogen response, as suggested by Reviewer 2. We have also expanded the introduction and discussion to emphasize the importance of this work. This includes statements that highlight the broader importance of this work in terms of plant responses to abiotic and biotic stresses: 

Line 64: “The pattern and degree of pectin methylation also impacts plant susceptibility to microbial infection [19], for instance wheat cultivars with more blockwise distribution of methyl esters were more susceptible to fungal infection than cultivars with more random methylation patterns [20]. Therefore, changes in esterification of cell walls and associated transport and metabolism of the released methanol and acetic acid could provide a rapid mechanism for plants to respond to abiotic and biotic stress.”

Line 73: “MeOH production in plants is largely attributed to changes in chemical and physical cell wall properties associated with the hydrolysis of methyl esters of cell wall carbohydrates like pectin [11,27–29]. However, this assertion lacks experimental evidence, which we aim to address in the present study.”

Line 101: “Therefore, it is vital to quantify the relationships between cell wall esters and foliar meOH and AA emissions to evaluate the hypothesis that emissions derive from cell wall de-esterification and to understand their physiological and biochemical roles during plant growth and development, adaptation to abiotic and biotic stress, mortality, and biomass decomposition.” 

Line 396: “Therefore, in situ monitoring of atmospheric emissions of meOH and AA from terrestrial ecosystems could help improve predictions of both tree growth and mortality mechanisms and their sensitivities to environmental change.”

Comment 2: Also, I did not see the data link that, I believe, PLOS One requires. If I missed it, I apologize. If it's not yet in the ms., it should be added.

Response 2: We now include a link to download the data within the supporting information section: 

Line 585: “Supporting material consisting of the raw experimental data files collected and analyzed in this study are available in electronic form free of charge: accessed through Mendeley Data (http://dx.doi.org/10.17632/7bdwbwy6wn.1). The supplementary data (Size: 169 MB) includes raw methanol and acetic acid emission data obtained from the PTR-MS and online TD-GC-MS as well as raw colorimetric assay data for leaf bulk methyl and O-acetyl ester content organized as follows:

Volatile emission folder:

• Real-time time meOH and AA emission data during leaf desiccation experiments (PTR-MS) 

• Near real-time time meOH and AA emission data during leaf desiccation experiments (online GC-MS)

Cell wall esterification folder:

• Raw absorbance data and derived total methyl ester content of AIR samples (methyl ester assays)

• Raw absorbance data and derived total acetate ester content of AIR samples (acetate ester assays)

• Derived and compiled methyl and acetate ester content of AIR samples”

Reviewer 2

Comment 1: The manuscript “Cell wall O-acetyl and methyl esterification patterns of leaves reflected in atmospheric emission signatures of acetic acid and methanol” regards to the quantification of total meOH and AA emissions in Populus trichocarpa desiccated leaves at different developmental stages were taking into account the cell wall acetylation and methylation content highlighting the quantification of AA/meOH emission ratios as a potential tool for rapid phenotype screening of structural carbohydrate esterification. I consider the manuscript very interesting and topical in the field of plant physiography and biofuels. Developing a method based on two experimental approaches, able to dynamically quantify acetic acid simultaneously with methanol is very innovative. I find the presentation of the work quite good and I don't find too many problems in the language. However, there are some aspects that need to be reviewed before the work can be presentable.

Response 1: We greatly appreciate reviewer 2 for the review and support of our innovative approaches to study cell wall acetylation and methylation through simultaneous quantification of leaf cell wall esterification and atmospheric AA/meOH emission ratios. 

Comment 2: The introduction should be reviewed trying to replace some slightly dated references (Micheli et al etc). The role of PME, pectin methylesterification and methanol release in defense against pathogens should be described a little more thoroughly. For references see papers of these authors Bethke, Lionetti, Dorokhov.

Response 2: We have updated some references in the introduction; replacing Micheli et al (2001) with Wormit et al, International Journal of Molecular Sciences (2018), Dorokhov et al, Frontiers in Plant Science (2018) and Saffer, Journal of Integrative Plant Biology (2018) (ref numbers 30, 32, 33). We have also included more detailed explanations of pectin methylation and methanol release in relation to pathogen defense (Line 64; and Line 87), including references (ref numbers 19, 20, 37: Lionetti et al, Journal of Plant Physiology (2012), Weitholter et al, Molecular Plant Microbe Interactions (2003), Dorokhov et al, PLOS Pathology (2012).

Comment 3: The sentence on page 5 lane 106 suggests that the AA is expected to shrink during development ... review

Response 3: We have included some further clarification and references in the sentence to explain why we suggest that AA emissions are hypothesized to decrease with increasing leaf age. 

Line 123: “Given previous observations of decreasing foliar meOH emissions with leaf age [34], and the de-esterification of cell wall esters throughout plant development [48,49] we hypothesize that similar phenological pattern can be observed for AA emissions.” 

Comment 4: Should figures 2 in the abscissa axis be seconds or minutes?

Response 4: The abscissa axis is in seconds.

Comment 5: The statistical treatment of the data is not correctly presented. The histograms are not accompanied by any statistical sign (letters for ANOVA, asterisks, etc.). It is also unclear how many repetitions have been made. Every figure must have adequate references in this regard.

Response 5: We have updated the figures to include statistical signs where appropriate (figures 3 and 5). The figure legends also contain a statement “n=7” indicating the number of replicates that were included in each figure and statistical test.

Comment 6: Figure 6 is more to review. It must be moved to supplementary materials

Response 6: We believe that this figure is valuable for understanding the results by an inter-disciplinary audience likely unfamiliar with the structure and esterification patterns of plant carbohydrates, therefore we believe it is justified to leave this figure within the main article.

---

## [Editor Report · Decision Letter 1]

7 Apr 2020

Cell wall O-acetyl and methyl esterification patterns of leaves reflected in atmospheric emission signatures of acetic acid and methanol

PONE-D-19-34824R1

Dear Dr. Dewhirst,

I am pleased to inform you that your revised manuscript has successfully addressed the reviewers' comments and it will be formally accepted for publication once it complies with all outstanding technical requirements.

With kind regards,

Federico Brilli, Ph.D.

Academic Editor

PLOS ONE
---

## [Editor Report · Acceptance letter]

8 May 2020

PONE-D-19-34824R1 

Cell wall O-acetyl and methyl esterification patterns of leaves reflected in atmospheric emission signatures of acetic acid and methanol 

Dear Dr. Dewhirst:

I am pleased to inform you that your manuscript has been deemed suitable for publication in PLOS ONE. Congratulations! Your manuscript is now with our production department. 

With kind regards,

on behalf of

Dr. Federico Brilli 

Academic Editor

PLOS ONE